# Metabolic Interplay in the Tumor Microenvironment: Implications for Immune Function and Anticancer Response

Reem Youssef [1], Rohan Maniar [2], Jaffar Khan [1] and Hector Mesa [1,*]

1 Department of Laboratory Medicine and Pathology, Indiana University School of Medicine, Indianapolis, IN 46202, USA
2 Division of Hematology/Oncology, Department of Medicine, Indiana University School of Medicine, Indianapolis, IN 46202, USA
* Correspondence: hmesa@iu.edu

**Abstract:** Malignant tumors exhibit rapid growth and high metabolic rates, similar to embryonic stem cells, and depend on aerobic glycolysis, known as the "Warburg effect". This understanding has enabled the use of radiolabeled glucose analogs in tumor staging and therapeutic response assessment via PET scans. Traditional treatments like chemotherapy and radiotherapy target rapidly dividing cells, causing significant toxicity. Despite immunotherapy's impact on solid tumor treatment, gaps remain, leading to research on cancer cell evasion of immune response and immune tolerance induction via interactions with the tumor microenvironment (TME). The TME, consisting of immune cells, fibroblasts, vessels, and the extracellular matrix, regulates tumor progression and therapy responses. TME-targeted therapies aim to transform this environment from supporting tumor growth to impeding it and fostering an effective immune response. This review examines the metabolic disparities between immune cells and cancer cells, their impact on immune function and therapeutic targeting, the TME components, and the complex interplay between cancer cells and nontumoral cells. The success of TME-targeted therapies highlights their potential to achieve better cancer control or even a cure.

**Keywords:** tumor microenvironment; immunotherapy; oncometabolites

## 1. Introduction

Benign cells focus on specialization, while malignant cells prioritize replication, making specialization and replication mutually exclusive. In specialized tissues, regeneration is limited to tightly regulated stem cell niches with restricted differentiation abilities and localizations in every tissue [1]. These somatic stem cells give rise to most malignant neoplasms. Transformed stem cells lose spatial confinement but maintain stem cell properties such as self-renewal, resiliency, pluripotency, and migration; they no longer respond to regulatory mechanisms, and undergo rapid evolution to adapt to the changes induced by their own growth, host response, and therapies [2].

Tumor biologists have focused on unraveling the genetic and metabolic pathways that give malignant cells a growth and survival advantage. Therapies for unresectable malignancies primarily target rapidly dividing cells and are the basis of most chemotherapy and radiotherapy protocols. However, this approach comes with significant toxicity to normal stem cells. In some gender-specific cancers, it was found that tumor growth and progression were influenced by sex hormones, leading to the development of hormone deprivation therapies. [3]. Cytogenetics advances revealed that a subset of malignancies have recurring translocations producing predictable abnormal proteins that can be targeted with specific "targeted" therapies [2]. Recent molecular-genetic technologies showed that amid the numerous genetic aberrations in cancer cells, only a limited subset of "driver mutations", initiates and advances malignancies [4]. In the 20th century, environmental

carcinogens were identified through epidemiology. In the 21st century, advances in epigenetics began to unravel how the environment contributes to tumor development [5]. Collectively, advances in cytogenetics, molecular genetics, and epigenomics have led to a new era of targeted oncologic therapies that have benefited only a small number of patients. The concept of engaging the immune system against cancer cells began in the 19th century with bacterial inoculations to treat tumors and evolved with immunology breakthroughs in the 20th century. Vaccines from tumor lysates or cancer-specific peptides, monoclonal antibodies against specific tumor antigens, adoptive cell therapies, and immunocytokines have shown durable benefits in a small fraction of patients prompting research focused on decoding how cancer cells evade the immune response [6,7]. This review explores metabolic alterations of cancer cells affecting the tumor immune microenvironment and potential therapeutic targets.

## 2. Targeting the Tumor Microenvironment

The tumor microenvironment (TME) includes immune cells, fibroblasts, vessels, and the extracellular matrix, which play a crucial role in regulating tumor progression and therapy response. Immuno-oncology revolutionized cancer treatment by using the immune system to detect and destroy cancer cells. Although the response to immunotherapy is delayed compared to other treatments, it can produce durable responses in advanced cancers for which, until recently, only palliation could be offered [8]. Immuno-oncology employs four primary strategies: immune checkpoint inhibitors, adoptive cell therapy, immunocytokines, and cancer vaccines. Immune checkpoint inhibitors target proteins that normally inhibit immune cells from attacking cancer cells, including PD-1/PD-L1, CTLA4, and LAG-3 pathways. Adoptive cell therapy enhances the inherent capabilities of effector T-cells and NK-cells to combat cancer and encompasses chimeric antigen receptor T cells (CAR-T), tumor-infiltrating lymphocytes (TIL), engineered T cell receptors (TCR), and NK-cell infusions [9]. Immunocytokines are endogenous small molecules that modulate immune responses through the regulation of proliferation and differentiation. While effective in promoting antitumor immune activity their usage is marred by significant toxicity and inconvenient administration [10]. Tumor vaccines have shown limited efficacy in treating cancer historically [11]. Advances in neoantigen target identification and novel delivery platforms bring renewed hope, yet their full potential remains unexplored [12]. Targeting tumoral angiogenesis with antiangiogenic agents has become a standard of care for certain malignancies. Yet, their effectiveness is limited by the emergence of tumor resistance and the risk of cardiovascular toxicity [13]. The success of therapies targeting TME emphasizes their significant potential for improving outcomes.

## 3. Divergent Metabolic Profiles in Cancer and Immune Cells: Implications for Therapy

### 3.1. Glucose Metabolism
3.1.1. Cancer Cells

The metabolism of high-grade malignancies resembles that of embryonic stem cells, relying more on glycolysis than oxidative phosphorylation to support proliferation and self-renewal. This preferential use of aerobic glycolysis is known as the "Warburg effect" [14]. Although glycolysis yields a lower production of ATP than the citric acid cycle (CAC) and oxidative phosphorylation (OXPHOS), it allows higher levels of glucose uptake because it occurs in the cytoplasm but not in the mitochondria, and it prevents the loss of carbon atoms as $CO_2$, so that they can be reused in other biosynthetic pathways [15]. The NADH-dependent reduction of pyruvate to lactate by the lactate dehydrogenase recycles the NAD, which is needed to sustain glycolysis. Pyruvate generated by glycolysis can enter the CAC through pyruvate dehydrogenase and carboxylase, yielding higher levels of energy and producing additional metabolic intermediates. Lactate released in the TME by the tumor cells reaches the circulation and can be converted back to pyruvate by normal cells and recaptured by tumor cells to further feed the CAC [16]. Bidirectional monocarboxylate transporters (MCTs) resulting in lactate import can also support tumor growth [17]. MCT1

and MCT4 expression has been frequently associated with poor outcomes across several cancers, including head and neck squamous cell carcinoma, nonsmall cell lung cancer, and melanoma [18–20]. Metabolic reprogramming of tumor cells leads to increased expression of glucose transporters and glycolytic enzymes through activation of various oncogenes and signaling pathways common to several malignancies. The glucose transporter family is expressed on the membrane of nearly all cells, with overexpression of glucose transporter 1 (GLUT1) seen in numerous tumors and correlating with rapid proliferation [21,22].

### 3.1.2. Immune Cells

Glucose utilization in immune cells is more regulated and more flexible. Resting cells rely on OXPHOS, which is more efficient in terms of ATP production. Activated immune cells can switch to glycolysis to meet increased energy demands; however. malignant cells outcompete immune cells for glucose and oxygen uptake, and the increased production of lactate creates a hypoxic and acidotic TME that weakens the activation of immune cells and polarizes the cellular responses towards immune anergy [14,15]. Additionally, the presence of immune checkpoints in the TME has been associated with T cell metabolic dysfunction, including impaired mitochondrial ATP production limiting T cell self-renewal despite a glycolytic phenotype, and promoting terminal differentiation [23]. The utilization of limited supplies of glucose within the TME is further complicated by the presence of protumor immune cells, including myeloid-derived suppressor cells (MDSCs). Tumor-associated monocytic MDSCs are characterized by their prominent utilization of glucose in the TME and generation of by-products that inhibit reactive oxygen species (ROS)-mediated apoptosis and promote polarization of early myeloid-derived cells to immunosuppressive phenotypes [24]. Enhancement of GLUT1 expression on tumor-associated neutrophils is associated with increased cell survival and tumor-supportive properties, highlighting the shared impact of metabolic reprogramming on the tumor and TME [25]. Additionally, the effect of aberrant glucose metabolism in TME can be seen in the intrinsic regulation of tumor-associated macrophages resulting in epigenetic remodeling and signal transduction that promotes an immunosuppressive phenotype [26]. Dendritic cells (DCs) play a crucial role in the activation of the adaptive immune response and their function is heavily influenced by glucose metabolism. Restricted glycolysis coupled with increased lactate can inhibit DC activation resulting in diminished antigen presentation, cytokine production, and T cell stimulation [27]. Limited availability of glucose may also impair CCR7 oligomerization required for cytoskeletal remodeling and diminish DC trafficking to tumor-draining lymph nodes diminishing immune cell mobilization [28].

### 3.1.3. Therapeutic Targeting

The Warburg effect is the basis for using radiolabeled glucose analogs to stage, survey, and assess therapeutic response of malignancies through positron emission tomography (PET) scans (Figure 1). Numerous drugs that inhibit glycolysis, glucose, glutamine, and lactate metabolism are being investigated; however, their efficacy has been limited by the numerous isoforms of metabolic enzymes, metabolic heterogeneity of tumor subpopulations, alternative pathways for energy production, and off-target side effects [29].

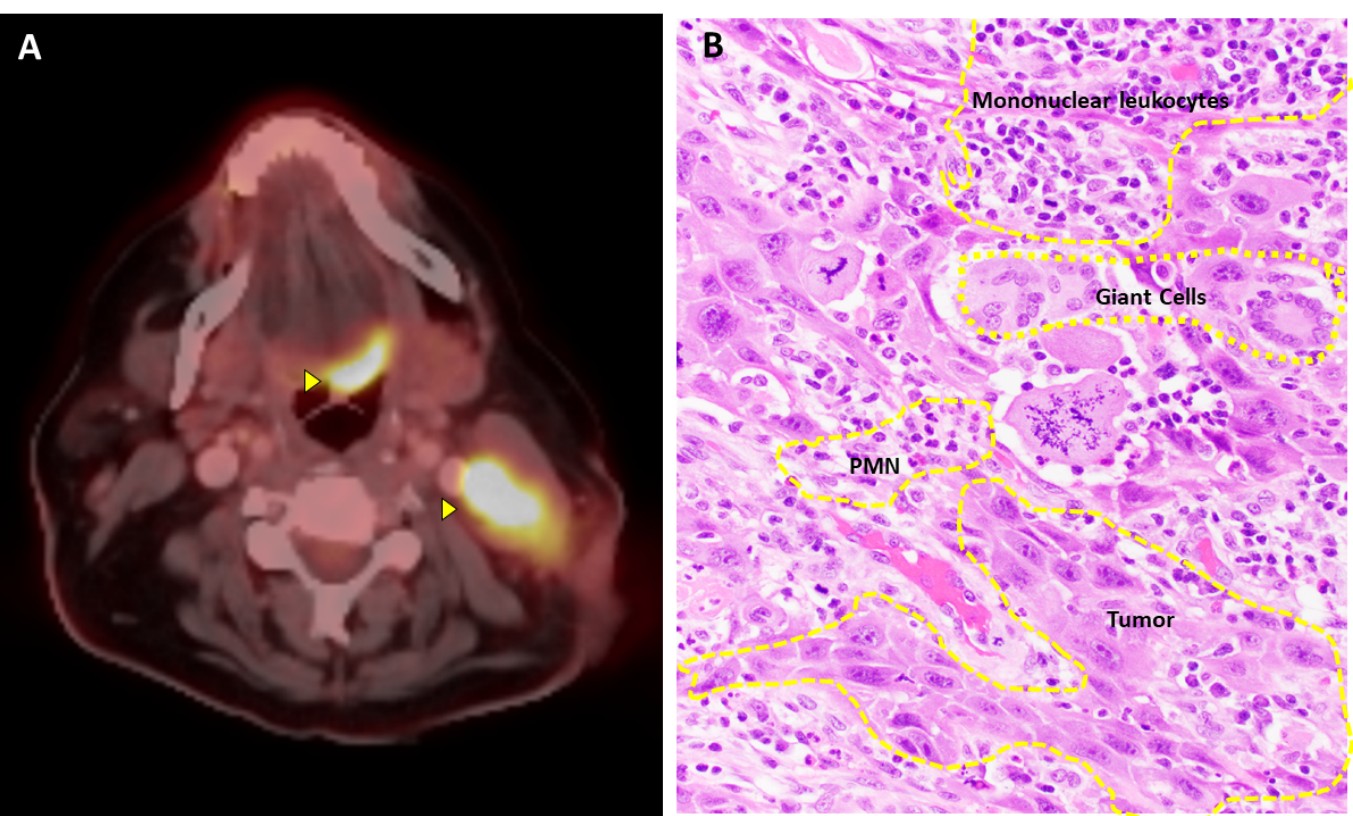

**Figure 1. Tumor-immune microenvironment.** (**A**) **Warburg effect:** Positron emission tomography visualizes a tongue base carcinoma (upper arrowhead) and neck lymph node metastasis (lower arrowhead) using radioactive glucose analogs. (**B**) **Cancer-associated inflammation:** The image exhibits mononuclear leukocytes (lymphocytes, histiocytes, plasma cells), multinucleated giant cells, and polymorphonuclear leukocytes (PMN) closely intertwined with tumor cells. Despite the prevalent presence of inflammatory cells in most tumors, the immune response tends toward immune tolerance (Hematoxylin and Eosin, 40× magnification).

### 3.2. Amino Acid Metabolism

3.2.1. Cancer Cells

High-grade malignancies have increased amino acid requirements for their rapid proliferation. They often upregulate amino acid transporters to acquire them from the TME and use the pyruvate generated through glycolysis or the recirculation of lactate to support the CAC, which provides substrates for de novo amino acid synthesis [30]. Many tumors rely on glutaminolysis, resulting in the conversion of glutamine to glutamate, for energy and intermediate metabolite production, and some cancer cells lose their ability to synthesize specific nonessential amino acids, having to compensate with upregulation of specific transporters for their survival, creating dependencies that can be exploited through targeted therapies [30]. Overexpression of *c-Myc* can result in alterations in glutamine metabolism, promoting glutaminolysis via glutaminase 1 regulation and upregulation of high-affinity glutamine transporters [31,32]. The catabolism of branched-chain amino acids (BCAAs) like valine, leucine, and isoleucine, can result in glutamate formation through nitrogen donation to $\alpha$-ketoglutarate; accumulation of BCAAs has been shown to impact the mTORC1 signaling pathway resulting in tumor progression [33]. Arginine is a conditionally essential amino acid with metabolic dependency on extracellular sources associated with specific cancers, such as melanoma and liver cancer [34,35]. Arginine serves as a precursor to glutamate synthesis, nitric oxide formation, and activation of mTORC signaling [36].

### 3.2.2. Immune Cells

Immune cells also require amino acids for protein synthesis, proliferation, and cytokine production. Their amino acid uptake is influenced by their activation state; however, malignant cells outcompete immune cells causing a depletion of amino acids in the TME, which contributes to a blunted immune response [37]. Amino acid metabolism plays a crucial role in T cell biology, including activation and clonal expansion through Slc7a5 interactions with mTORC1 and differentiation into effector and memory T cells through epigenetic modulation [37]. Deprivation of essential amino acids by MDSCs through arginase1 and nitric oxide synthase 1 and 2 activity also regulates T cell function, reducing the formation of memory T cells, and promoting the release of peroxynitrite, which can induce T cell apoptosis [38–41]. The induction of *indoleamine-pyrrole 2,3 dioxygenase* (*IDO-1*) on both myeloid and tumor cells results in decreased levels of tryptophan and the presence of immunosuppressive catabolites such as kynurenine (Kyn) that promote Treg activity and diminish T cell effector function [42–44]. The broad impact of arginine and tryptophan catabolism and the formation of potent metabolites can also be seen on DCs, which are similarly impacted by the presence of Kyn, resulting in downstream activation of the aryl hydrocarbon receptor creating a positive feedback loop of *IDO-1* activation [27,45,46]. The importance of IDO-1 as a modulator of immune cell activity resulted in the development of inhibitors across several solid tumors with numerous preclinical models and early-phase drug testing demonstrating promising results [47]. Unfortunately, the clinical development of these drugs faced disappointing results in large clinical trials, highlighted by the failure of the phase 3 ECHO-301/KEYNOTE-252 trial to show a significant benefit with the addition of an IDO-1 inhibitor to pembrolizumab, in the treatment of advanced melanoma [48]. A follow-up analysis evaluated IDO-1 expression via immunohistochemistry in primary and metastatic melanoma tumors and noted significant heterogeneity in IDO-1 expression across longitudinal samples potentially explaining these results [49].

### 3.2.3. Therapeutic Targeting

Targeting amino acid metabolism in cancer cells has shown success in a small subset of malignancies, for example, the use of asparaginase in lymphoblastic leukemias and PEGy-lated arginine deaminase in hepatocellular carcinoma and mesothelioma. Treatment with glutamine inhibitors and recombinant methioninase has shown promising results in animal models or early clinical studies [47]. Deprivation therapies against nonessential amino acids should provide a competitive advantage to immune cells because they retain the capacity to synthesize these compounds and have shown synergistic effects with immunotherapy in animal models [50]. Despite the described setbacks in phase 3 clinical trials, ongoing studies continue to look at the role of IDO-1 inhibitors while honing predictive biomarkers, patient selection, and combinatorial strategies to enhance efficacy [47] (Table 1).

**Table 1.** Critical disparities in metabolic processes between normal and tumor cells.

| Metabolite | Immune Cells | Malignant Cells | Net Effect |
|---|---|---|---|
| Glucose | Regulated uptake Resting cells: oxidative phosphorylation Activated cells: glycolysis | Dysregulated uptake Glycolysis → Lactate, hypoxia, acidosis | Malignant cells outcompete immune cells; acidotic environment leads to impaired immune function |
| Amino acids | Regulated, influenced by activation and differentiation status | Dysregulated, upregulation of specific pathways, e.g., glutamine → tricarboxylic acid cycle | Tumor-induced amino acid depletion leads to impaired immune function |
| Nucleotides | Synthesis is tightly regulated, and switch between de novo and salvage/recycling pathways | Upregulation of de novo synthesis pathways over recycling leads to depletion of local resources | Nucleotide depletion leads to impaired immune activation and proliferation |

**Table 1.** *Cont.*

| Metabolite | Immune Cells | Malignant Cells | Net Effect |
|---|---|---|---|
| Fatty acids | Fatty acid transporters are tightly regulated, skewed towards lipolysis | Upregulation of fatty acid transporters and de novo synthesis for energy and membrane assembly | Depletion of fatty acids leads to impaired immune activation and proliferation |
| Cholesterol | Regulated, influenced by activation and differentiation status | Deregulation of synthesis and uptake and reduced degradation allow proliferation and signaling | Dysregulation of metabolism affects the expression of immune checkpoints, polarizes response toward immunosuppression, increases pro-oncogenic cytokines |
| Oxygen | Highly dependent on oxygen for the generation of reactive oxygen species and production of cytokines | High resistance to hypoxia through activation of hypoxia-inducible factors and glycolysis | Hypoxia hinders the movement and function of immune cells—favors the function of regulatory over effector T cells |

### 3.3. Nucleotide Metabolism

3.3.1. Cancer Cells

High-grade malignant cells have very high nucleotide requirements. Nucleotides are essential for nucleic acid synthesis, enzyme regulation, and metabolism. Cancer cells rely more on de novo synthesis than external sources, a process that requires large amounts of energy that leads to the depletion of nucleotide precursors in the TME [51]. The de novo synthesis by cancer cells involves upregulation of the numerous enzymatic pathways required for the synthesis, modification, and assembly of precursors such as ribose for purines, deoxyribose for pyrimidines, $CO_2$, amino acids, and tetrahydrofolate [51]. Under normal circumstances, the synthesis of pyrimidines is simpler than that of purines; purines exert an inhibitory effect over the enzymes participating in purine synthesis, but activate enzymes needed for pyrimidine formation and vice versa [52]. However, in cancer cells, these processes are upregulated and dysregulated, usually due to the abnormal function of oncogenes (e.g., *K-RAS*, *c-MYC*), tumor suppressor genes (e.g., *p53*, *RB1*, *P16*$^{INK4A}$) or genes that have dual oncogenic and tumor suppressor functions (e.g., *ATF3*) leading to genomic instability and tumor progression [53,54].

3.3.2. Immune Cells

Immune cells also require nucleotides for replication, but in contrast to malignant cells, they can switch between de novo synthesis and salvage pathways. In the salvage pathway, bases and nucleosides resulting from the degradation of nucleic acids are recycled [55]. In general, immune cells are more efficient, use less energy, and are less dependent on precursors from the TME, making immune cells less vulnerable to antimetabolite therapies. However, dysregulated nucleotide metabolism affects the immune response through Toll-like, RIG-like, NOD-like, purinergic, and adenosine receptors. In many tumors, but not all, the net effect is inhibitory through the recruitment and expansion of suppressor Tregs [56]. The adenosine pathway and its implication on TIME and immune cell composition have been an area of interest for therapeutic development. The accumulation of adenosine in the TME, produced by CD39 and CD73 and driven by the downregulation of intracellular transport in the setting of hypoxia, creates a pro-angiogenic effect with modulation of the local immune system toward anergy [57]. The adenosine receptors A1, A2A, A2B, and A3 are ubiquitously expressed on myeloid cells and lymphocytes and appear to primarily attenuate the immune response via inactivation of TNF-α production, augmentation of IL-10 production, suppression of IL-2 secretion, and upregulation of immune checkpoints such as CTLA4 and PD1 [58].

### 3.3.3. Therapeutic Targeting

Nucleoside analogs and folate antagonists comprise a large proportion of the standard chemotherapy regimens and have been part of the oncologic armamentarium for a long time; while effective, they are associated with significant toxicity to the stem cells of tissues with high turnover rates. Newer therapies targeting nucleotide metabolism include highly specific inhibitors of specific enzymes downstream from driver mutations in different types of cancer. Most of these agents are being tested in phase I/II clinical trials. Several of these agents have shown synergistic effects when combined with immunotherapy in animal models [59]. Targeting nucleotide metabolism should also produce imbalances in the purine/pyrimidine ratios, leading to an increased tumor mutational burden (TMB), which results in increased expression of surface neoantigens and thus augments the effectiveness of immunotherapy [60]. Given the potential synergy of adenosine-directed therapies with immunotherapy, several studies have sought to evaluate the role of combination therapy in the treatment of cancer, as summarized in Table 2.

### 3.4. Fatty Acid Metabolism

#### 3.4.1. Cancer Cells

High-grade cancer cells have an increased demand on fatty acids for energy, membrane synthesis, and the generation of signaling intermediates, such as eicosanoids, fatty acid carnitines, thioesters, and N-acyl ethanolamines [61]. Fatty acids also modulate the function of proteins involved in cell growth, proliferation, motility, and survival through the PI3K/AKT/mTOR and RAS/RAF/MEK/ERK pathways [62]. High-grade cancer cells favor de novo synthesis from nonlipid substrates, a process in which fatty acids are generated from simple precursors predominantly in the cytosol, and, to a lesser extent, in the mitochondria [63]. Additionally, they also upregulate fatty acid transporters to capture fatty acids from the circulation or the TME through passive diffusion or transport proteins leading to their depletion in the TME [64]. Lipid droplets are common in many types of tumors and serve as energy reserves through lipolysis and lipophagy. A metabolic shift towards fatty acid oxidation appears to be a common mechanism of therapy resistance [64].

Recent studies on dietary interventions highlighted the crucial role of saturated and unsaturated fatty acid ratios in impacting cancer cell survival and their resistance to therapy. This balance significantly influences their ability to endure oxidative stress and undergoing ferroptosis, key processes in cancer treatment resistance [65]. Among these, the ratio of the essential polyunsaturated fatty acids (PUFA) omega-3 ($\omega$-3) and omega-6 ($\omega$-6) has been shown to affect the expression of signaling molecules altering downstream factors and pathways. An increased $\omega$-3/$\omega$-6 ratio results in a reduction of the production of pro-inflammatory and procarcinogenic $\omega$-6 derivatives [65].

Oleic acid (OA), a $\omega$-9 fatty acid, promotes the proliferation of breast cancer cells by activating specific signaling pathways linked to G protein-coupled receptors (GPR) 40 and 120 [66,67] and a network of signaling pathways involving PKC, ERK, EGFR, MMP, PI3K/Akt, and PLD2/mTOR [68,69].

Stearoyl-CoA desaturase-1 (SCD1), the limiting enzyme for monounsaturated fatty acid synthesis, also shows a robust correlation with cancer development across various cancer types. The PI3K/Akt/mTOR signaling pathway amplifies SCD1 expression, contributing to lipogenesis in cancer cells. Additionally, the tumor suppressor p53 directly targets SCD1, supporting the idea that increased SCD1 expression and activity play a critical role in cancer development [68,69], making it a potential therapeutic target.

#### 3.4.2. Immune Cells

Immune cells also require fatty acids for similar purposes. However, the uptake of these fatty acids is regulated and influenced by their activation state. Activated immune cells may shift toward increased fatty acid oxidation to meet their energy needs. Insufficient fatty acids in the TME can impair an effective antitumor response [70]. De novo fatty acid synthesis remains an integral part of T cell activation coupled with the shift towards the

glycolytic pathway. T cell differentiation appears to be highly coupled to intracellular programs of lipid metabolism and extracellular lipid exposure [70,71]. Similarly, MDSC and DC differentiation and function are coupled with lipid metabolism; the accumulation of oxidized fatty acids leads to augmented suppressive functions of polymorphonuclear, MDSCs, and DC dysfunction through impaired antigen presentation and immune tolerance [72–74].

### 3.4.3. Therapeutic Targeting

Inhibitors of enzymes involved in fatty acid synthesis or fatty acid transporters are currently under investigation as potential cancer treatments. However, research in this area is still in its early stages. Enzymes such as fatty acid synthase (FASN), acyl-CoA thioesterases (ACOTs), and acetyl CoA carboxylase (ACC), which play a crucial role in the de novo synthesis of fatty acids, have been targeted, resulting in restricted growth, proliferation, and metastasis in both cancer cell lines and animal models [75,76]. The effect of dietary interventions to alter the ratios of essential fatty acids such as $\omega$-3 and $\omega$-6 PUFAs and $\omega$-9 MUFAs for cancer prevention due to their anti- or pro-inflammatory/carcinogenic effects has garnered recent attention. [77,78]. Unfortunately, a recent extensive systematic review showed that dietary interventions using MUFA/PUFAs have a negligible effect on cancer prevention, treatment, or death [79].

### 3.5. Cholesterol Metabolism

#### 3.5.1. Cancer Cells

Cancer cells undergo metabolic reprogramming to sustain increased synthesis or uptake of cholesterol needed for the biosynthesis of organelles and plasma membranes. In some cancers, cholesterol metabolism is involved in sustaining oncogenic signaling through the Hedgehog−Smoothened or the mammalian target of rapamycin complex 1 (mTORC1) pathways [80–82]. Upregulation of Sterol Regulatory Element-Binding Protein 2 (SREBP2) via the AKT–PCK1–INSIG1/2–SREBP axis and RAR-related orphan receptor gamma (ROR$\gamma$) are common mechanisms of cancer cells to upregulate de novo synthesis of cholesterol. Additionally, cholesterol levels affect the ability of tumor cells to withstand oxidative stress and undergo ferroptosis [83–85].

#### 3.5.2. Immune Cells

Cholesterol and its metabolites act as signaling mediators and are involved in both innate and adaptive reactions [86]. Increased cholesterol levels in the TME induce the expression of immune checkpoints and exhaustion in CD8+ T cells in animal models, therefore impairing antitumor immune responses [87].

#### 3.5.3. Therapeutic Targeting

Cholesterol-lowering agents such as PCSK9 inhibitors have demonstrated potentiation of checkpoint inhibitors in preclinical studies [88]. The effects on tumor aggressiveness of mutated *TP53*, one of the most common genetic alterations in cancer, have been shown to be significantly linked to the regulation of cholesterol biosynthesis in breast cancer cell lines. P53-deficient cancer cells activate the mevalonate pathway via SREBP2 to reduce oxidative stress and promote the synthesis of pyrimidines. Inhibition of the mevalonate pathway with statins has shown promise in treating P53-mutated cancers [89].

### 3.6. Oxygen Dependency

#### 3.6.1. Cancer Cells

High-grade cancer cells frequently exhibit elevated hypoxia tolerance. They achieve this by promoting the development of abnormal vascular networks (tumoral neoangiogenesis), which slow down blood flow, induce hypoxia, impede the delivery of intravenous chemotherapy drugs, and enhance the entry of tumor cells into the bloodstream [90]. Cancer stem cells and their offspring capable of adapting to low-oxygen environments through the activation of hypoxia-inducible factors (HIF-1$\alpha$, HIF-2$\alpha$, HIF-1$\beta$) exhibit more

aggressive phenotypes compared to those that cannot adapt to hypoxia and undergo cell death. Hypoxia-inducible factor proteins regulate a multitude of genes and intricate RNA networks, encompassing those responsible for redirecting metabolism toward anaerobic pathways such as glycolysis or fatty acid oxidation for energy production, as discussed earlier. [90] Additionally, HIF proteins govern pathways associated with angiogenesis, cell survival, and invasive behavior [91]. Hypoxia promotes a stem-like phenotype in tumor cells through the activation of genes such as *OCT4*, *SOX2*, *c-MYC*, *CD44*, *CD133*, *WNT*, *Notch*, and *NANOG*, leading to dedifferentiated and undifferentiated tumor phenotypes associated with more aggressive biology [91]. Hypoxia has recently been recognized for its ability to induce epithelial-mesenchymal-transition (EMT) in epithelial cells. This process is driven by the activation of genes such as *SNAIL*, *ZEB1*, *TWIST*, *TCF3*, and *NF-κβ*, mediated by HIF-1α and other mechanisms. EMT entails the activation of signaling pathways that enable epithelial cells to adopt characteristics similar to stromal cells involved in reparative/regenerative processes. These characteristics include rapid growth, migration, angiogenesis induction, tissue remodeling, and the capacity to trigger an inflammatory response. EMT is linked to dedifferentiated and sarcomatoid carcinoma phenotypes, heightened local tumor aggressiveness, and increased metastatic potential [92].

### 3.6.2. Immune Cells

Effector immune cells require oxygen for the generation of reactive oxygen species (ROS) and the production of cytokines for mounting a robust immune response. ROS are particularly important for innate immune responses such as respiratory bursts and inflammasome activation [93]. In a hypoxic TME, there is an accumulation of lactic acid and adenosine due to the Warburg effect. The previously described A2A receptor (A2AR) can hinder the function and proliferation of effector cells by interacting with high levels of adenosine in the TME, initiating a series of intracellular events that diminish the responsiveness of effector cells to IL-2 through an mTOR pathway blockade [94]. Furthermore, within hypoxic TMEs, there is an elevation in the levels of chemokines CCL28 and CCL2 produced by tumor cells. These chemokines play a role in recruiting Tregs and macrophages. Tregs promote angiogenesis and suppress effector immune cells and antigen-presenting cells through both contact-dependent and independent mechanisms, such as the secretion of IL-10 and TGF-β. This contributes to immune evasion within the TIME [91,95].

The combination of a hypoxic and acidotic TME along with an expanded population of Tregs has been linked to the reprogramming of macrophages within the TME, skewing tumor-associated macrophages toward a M2 phenotype. M2 macrophages release cytokines such as IL-10, IL-1β, TGF-β, and VEGF, which exhibit anti-inflammatory, immunosuppressive, and pro-angiogenic properties. They also secrete matrix metalloproteins that contribute to the remodeling of the tumor stroma towards fibrosis (desmoplasia), further hindering the recruitment of effector cells and the diffusion of chemotherapeutic agents. This combined effect of Tregs and M2 macrophages leads to the inhibition of CD4+ helper cells, CD8+ cytotoxic T cells, increased tumoral angiogenesis, stromal activation, and remodeling and is considered one of the most critical factors in tumor-immune evasion and progression [96,97].

### 3.6.3. Therapeutic Targeting

Therapeutic strategies aimed at modifying the hypoxic TME include antiangiogenic agents (e.g., bevacizumab, ramucirumab, trebananib), HIF inhibitors (belzutifan, 6RK73), hypoxia-activated/bioreductive prodrugs (e.g., tirapazamine, evofosfamide, apaziquone) and less commonly, hyperbaric medicine. These agents are currently undergoing preclinical and clinical development, with some having received approval for specific clinical scenarios, typically in combination with traditional chemotherapy and immunotherapy regimens [97] (Table 2).

**Table 2.** Pharmacological agents targeting the TME.

| Metabolic Pathway | Mechanism of Action | Pharmacological Agent | Stage of Development |
|---|---|---|---|
| Glucose | GLUT, glucose transporter inhibitor | WZB117 | Preclinical [98] |
| | | BAY-876 | Preclinical [99] |
| | Hexokinase inhibitor | 3-Bromopyruvic acid (3-BrPA) | Preclinical [100] |
| | | Lonidamine (LND) | Phase II [101] |
| | | 2-Deoxy-d-glucose (2-DG) | Phase I [102] |
| Amino acid | Selective glutaminase inhibitor | Telaglenastat | Phase II [103] |
| | BCAT1 inhibitor | Gabapentin | Preclinical [104] |
| Nucleotide | Targeting purine or pyrimidine pathways | CAD blockage: Afatinib | FDA approved [105] |
| | DNA synthesis blockage | Capecitabine | FDA approved [106] |
| | | Decitabine | FDA approved |
| | Adenosine pathway inhibitors | Oleclumab | Phase II [107] |
| | | Dalutrafusp alfa | |
| | | HLX23 | ~50 phase I/II trials [108] |
| | | Anti CD73 antibodies | |
| Fatty acid | Acetyl-CoA carboxylase (ACC) inhibitor | 5-tetradecyloxy-2-furancarboxylic acid (TOFA) | Preclinical [109] |
| | ATP-citrate lyase (ACLY) inhibitors | GSK165 | Preclinical [110] |
| Cholesterol | Depletion of cholesterol hinders signaling and induces apoptosis | Statins | Phase II/III [111] |
| Hypoxia | Antiangiogenic agents | Ramucirumab | FDA approved [112] |
| | | Bevacizumab | FDA approved [113] |
| | | Trebananib | Phase II [114] |
| | HIF inhibitors | Belzutifan | FDA approved [115] |
| | | 6RK73 | Preclinical [116] |
| | | DFF332 | Phase I [117] |
| | | RO7070179 | Phase I [118] |
| | | NKT2152 | Phase II [119] |
| | Hypoxia-activated/bioreductive prodrugs | Tirapazamine | Phase II [120] |
| | | Evofosfamide | Phase II [121] |
| | | Apaziquone | Phase III |

## 4. Conclusions

The realization of the importance of nontumoral TME for the survival and propagation of cancer cells has been one of the most important breakthroughs in the understanding of tumor biology. The stem-cell-like properties of aggressive cancer cells allow them to reprogram nontumoral cells in the TME to align them with their goal of favoring replication to the detriment of specialization. Oncologic therapies have traditionally focused on exploiting vulnerabilities of tumor cells and remain the mainstay of oncologic treatment. The recent development of therapies targeting the nontumoral TME has yielded some of the most significant advances in recent years. In our review, we described some of the advances in the understanding of the incredibly complex crosstalk between cancer cells

and nontumoral immune cells in the TME, focusing on areas that appear promising for enhancing immunotherapies.

**Author Contributions:** Conceptualization, all authors; writing—original draft preparation, R.Y. and H.M.; writing—review and editing, R.M. and J.K.; supervision, R.M. and H.M.; project administration, H.M. All authors have read and agreed to the published version of the manuscript.

**Funding:** This research received no external funding.

**Acknowledgments:** The Department of Pathology and Laboratory Medicine of Indiana University School of Medicine, Indianapolis IN, provided support for this work.

**Conflicts of Interest:** The authors declare no conflict of interest.

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
