# Peer review of "Metabolic Interplay in the Tumor Microenvironment: Implications for Immune Function and Anticancer Response"

_cimb, doi:10.3390/cimb45120609_

Round 1

Reviewer 1 Report

Comments and Suggestions for Authors

The authors review some metabolic alterations in tumor cells that affect the tumor microenvironment and the immune response, comparing metabolic profiles between cancer cells and immune cells. The mansucript is well written and readable.

Even enlarging online the images B and E of Fig. 1 I could not identify what the arrows are supposed to indicate. It would be better if you included detailed inserts of these images which could show the items described in the legends.

Line 112 - Substitute "but not in" for "and not" 

Line 234 - Another example is p16INK4a which is mentioned by several authors, including Volgareva et al. 2004 (BMC Cancer 4: 58), Li et al. 2020 (J Cancer 11: 1457-1467), and Huo et al 2020 (J Clin Lab Anal 34: e23207)

Figure 1 - Write H&E in full  followed by respective abbreviation the first time it is mentioned

Author Response

We appreciate and thank the reviewers for their comments and suggestions. The edits, corrections and clarifications have been entered using a blue font.

Reviewer 1.

The authors review some metabolic alterations in tumor cells that affect the tumor microenvironment and the immune response, comparing metabolic profiles between cancer cells and immune cells. The mansucript is well written and readable.

Even enlarging online the images B and E of Fig. 1 I could not identify what the arrows are supposed to indicate. It would be better if you included detailed inserts of these images which could show the items described in the legends.

The figure has been simplified to reflect better the contents of the manuscript and now shows a more detailed depiction of the leukocytic subpopulations of the tumor immune microenvironment.

Line 112 - Substitute "but not in" for "and not" 

The substitution has been made.

Line 234 - Another example is p16INK4a which is mentioned by several authors, including Volgareva et al. 2004 (BMC Cancer 4: 58), Li et al. 2020 (J Cancer 11: 1457-1467), and Huo et al 2020 (J Clin Lab Anal 34: e23207)

p16INK4a with a corresponding reference has been added.

Figure 1 - Write H&E in full  followed by respective abbreviation the first time it is mentioned

The correction has been made.

Reviewer 2 Report

Comments and Suggestions for Authors

Dear authors,  

Thanks for your contribution to this field. This state of art on immune cells and cancer is intersting but quite frustrating since many elements are missing. The manuscript is well written but not well organized.  

First the chosen title doesn’t reflect the content of the manuscript. The introduction part is very long and out of focus to the subject. 

Since the manuscript is focused on immune component, it is better to choose the term “Tumor Immune microenvironment” to avoid mistake with the real TME which encompasses Fibroblasts/CAF and ECM. 

For each biomolecule, relations with cancer cells are presented but many important data are missing: lactate transporter for glucose, details on amino acid and some mutated enzyme in their metabolism like CD01, for lipids: fatty acid saturation status w3, w6, w9, data on cholesterol, metabolism, nucleotide: a reference is required lines 223-224. 

In figure 1, only part A is evidenced within the manuscript. It is quite difficult to understand the usefulness of the other parts. 

These points need to be solved to improve manuscript potential before considering its acceptance for publication. 

Regards,

Author Response

We appreciate and thank the reviewers for their comments and suggestions. The edits, corrections and clarifications have been entered using a blue font.

Reviewer 2.

Dear authors,  

Thanks for your contribution to this field. This state of art on immune cells and cancer is intersting but quite frustrating since many elements are missing.

We understand and share the reviewer's opinion. This is only one of several articles on a special issue dedicated to this very broad topic. We hope that the remaining articles can cover all the important subjects our focused review did not address. 

The manuscript is well written but not well organized.  

Minor adjustments have been made throughout the manuscript to highlight the article's focus and enhance the manuscript's overall organization and flow. Let us know if we accomplished this goal.

First the chosen title doesn’t reflect the content of the manuscript.

The manuscript title has been changed to: “Metabolic Interplay in the Tumor Microenvironment: Implications for Immune Function and Anti-Cancer Response”. We think the new title now reflects the contents of the manuscript.

The introduction part is very long and out of focus to the subject.

The introduction has been shortened from 592 words to 347. The purpose of the introduction is to show immunotherapy’s place in the history of oncologic therapies and how the focus of fighting cancer has shifted as the understanding of tumor biology has evolved, especially the realization that non-tumoral cells within tumors are not just mere bystanders. 

Since the manuscript is focused on immune component, it is better to choose the term “Tumor Immune microenvironment” to avoid mistake with the real TME which encompasses Fibroblasts/CAF and ECM. 

The correction has been applied across the manuscript wherever pertinent. However immune cells share the TME with all the other mentioned components.

For each biomolecule, relations with cancer cells are presented but many important data are missing:

  • lactate transporter for glucose

A new section discussing lactate transporter for glucose has been added.

  • details on amino acid and some mutated enzyme in their metabolism like CD01

We reviewed the literature, and we could not find anything about CD01 mutation. Is it possible this could be a typo and that the reviewer was referring to IDO-1? If that is the case, our manuscript already discusses extensively the role of IDO-1; the section has been highlighted in blue. We found literature about CD1 mutations, but this relates to neurodegenerative disease.

  • for lipids: fatty acid saturation status w3, w6, w9, data on cholesterol metabolism

A new section on omega 3, 6, 9 PUFA/MUFAs has been added.

A new section on cholesterol metabolism has been added.

  • nucleotide: a reference is required lines 223-224.

A reference has been added.

In figure 1, only part A is evidenced within the manuscript. It is quite difficult to understand the usefulness of the other parts. 

Agreed. The figure has been simplified and now shows a more detailed depiction of the tumor immune microenvironment.

These points need to be solved to improve manuscript potential before considering its acceptance for publication. 

Regards,

We appreciate the feedback. Please let us know if the edited version passes muster.

Round 2

Reviewer 2 Report

Comments and Suggestions for Authors

Dear authors,

Thanks for considering my comments.

The manuscript has been quite improve and, now, is suitable for publication.

Regads,